# Underwater Image Classification Algorithm Based on Convolutional Neural Network and Optimized Extreme Learning Machine

Junyi Yang [1] , Mudan Cai [2], Xingfan Yang [3] and Zhiyu Zhou [3],*

1    School of Mechanical Engineering, Hangzhou Dianzi University, Hangzhou 310018, China
2    School of Electronics and Information, Hangzhou Dianzi University, Hangzhou 310018, China
3    School of Computer Science and Technology, Zhejiang Sci-Tech University, Hangzhou 310018, China
*    Correspondence: zhouzhiyu1993@zstu.edu.cn

**Abstract:** In order to deal with the target recognition in the complex underwater environment, we carried out experimental research. This includes filtering noise in the feature extraction stage of underwater images rich in noise, or with complex backgrounds, and improving the accuracy of target classification in the recognition process. This paper discusses our contribution to improving the accuracy of underwater target classification. This paper proposes an underwater target classification algorithm based on the improved flow direction algorithm (FDA) and search agent strategy, which can simultaneously optimize the weight parameters, bias parameters, and super parameters of the extreme learning machine (ELM). As a new underwater target classifier, it replaces the full connection layer in the traditional classification network to build a classification network. In the first stage of the network, the DenseNet201 network pre-trained by ImageNet is used to extract features and reduce dimensions of underwater images. In the second stage, the optimized ELM classifier is trained and predicted. In order to weaken the uncertainty caused by the random input weight and offset of the introduced ELM, the fuzzy logic, chaos initialization, and multi population strategy-based flow direction algorithm (FCMFDA) is used to adjust the input weight and offset of the ELM and optimize the super parameters with the search agent strategy at the same time. We tested and verified the FCMFDA-ELM classifier on Fish4Knowledge and underwater robot professional competition 2018 (URPC 2018) datasets, and achieved 99.4% and 97.5% accuracy, respectively. The experimental analysis shows that the FCMFDA-ELM underwater image classifier proposed in this paper has a greater improvement in classification accuracy, stronger stability, and faster convergence. Finally, it can be embedded in the recognition process of underwater targets to improve the recognition performance and efficiency.

**Keywords:** underwater image classification; convolutional neural network; extreme learning machine; flow direction algorithm; chaos initialization; multiple population strategy; fuzzy logic

## 1. Introduction

The classification and identification of marine organisms such as fish, plankton and coral reefs are conducive to the management of marine biological systems and marine biodiversity and the analysis of marine biological species differences and the protection of endangered marine organisms. Studying the distribution of various marine organisms is helpful to analyze the impact of global warming and human exploitation of marine resources on marine organisms and to guide human rational exploitation of marine resources. However, underwater imaging has the characteristics of edge and detail degradation and low contrast between target and background and noise pollution due to the complexity of the underwater environment [1]. Conventional classification algorithms are difficult to distinguish important features and obtain effective information, which makes underwater image classification a very challenging task.

The difficulty of underwater image classification is to extract effective features from underwater images full of noise [2,3]. Early underwater image classification commonly used methods are based on image processing and pattern recognition technology, using filtering and other methods to preprocess underwater images or perform segmentation and other operations [4,5]. Spampinato calculated gray histogram for fish classification by capturing image features, such as contour shape and scale texture of underwater objects [6]. A method was proposed [7] to reduce the impact of various disturbances in underwater environment by training fish images with support vector machines (SVM). In the data set of 15 fish species with a total of 24,000 images, they improved the classification accuracy to 74.8%.

With the sudden emergence of artificial intelligence [8,9] and convolutional neural network (CNN) [10,11], many novel and efficient methods have been added to image classification. Villon proposed a deep learning classification method based on CNN [12] to identify fish in coral reefs. Salman compared the effects of a variety of traditional machine learning classification methods with convolutional neural network classification methods [13] on the fish data sets of LIFECLEF 2014 and LIFECLEF 2015 and achieved a correct classification rate of more than 90%.

The advantage of using CNN for image classification is that there is no need to manually extract and filter image features. Convolution operation can automatically complete this work. With the deepening of convolution, neural networks can produce higher semantic level features for classification. Qin et al. [14] designed the deepfish framework for the classification of marine fish and achieved 98.64% accuracy in the experiment of Fish4Knowledge dataset. Labao et al. [15] developed a set of fish recognition and detection system combined with long short-term memory network and convolution neural network based on region and tested it in 18 video data taken in the field to realize the function of fish recognition and detection.

For underwater target classification based on CNN, due to the cascade convolution of CNN, only high semantic feature information can be generated. In order to ensure the accuracy of target recognition and classification in complex underwater environments, all effective information should be fully applied. Besides the high semantic features, it is also necessary to make full use of low-level features such as texture and line, fish dorsal fin, fish scale texture [16], mouth line, etc. These can be extracted in the shallow convolution layer. Guo et al. [17] used the depth residual network to complete the identification of sea cucumber, with the highest accuracy of 89.53%. Prasetyo et al. [18] introduced the residual network into the CNN network and proposed a VGGNet with multi-level residual MLR-VGGNet. It retains the primary and intermediate features from the early convolution blocks and integrates the deep advanced features. The classification accuracy of MLR-VGGNet is 99.69% on FishGres and Fish4Knowledge datasets. In addition to introducing the residual network, Anabel et al. [19] constructed a two-level classifier using three CNN models to classify the structure, shape, and texture of coral, respectively. Ananda et al. [20] applied ResNet152 to the classification and detection of brain images after transfer learning. Furthermore, the attention mechanism is introduced into the network to train the network to assign weights to different features and to pay attention to more important features and to ignore secondary features [21–23].

However, these improvements still cannot get rid of the problem of uneven underwater image quality. Researchers began to introduce image data enhancement and other technologies into the classification work to further improve the performance. Tabik et al. [24] used ImageNet and MLC-2008 coral data sets to migrate CNN and analyzed the impact of data enhancement, including a variety of artificial distortions to increase the volume of the data set, such as brightness adjustment, scaling, and rotation, on the classification accuracy. Dutta et al. [25] proposed a fish quality analysis technology based on image processing, which achieved 95% to 100% classification sensitivity. Alshdaifat et al. [26] corrected the brightness of underwater fish video to remove blur and used the example segmentation method to obtain 95.2% accuracy on Fish4Knowledge dataset.

In order to further improve the classification accuracy of underwater images, the extreme learning machine (ELM) is used to classify features instead of softmax classifier commonly used in CNN [27–31]. In order to reduce the large amount of training time spent by traditional CNN using back propagation (BP) mechanism, Huang et al. [32] published a feedforward neural network (FNN) called the extreme learning machine (ELM), which improves the learning efficiency of neural networks and simplifies the setting of parameters. The random initialization of feature mapping from input layer to output layer in ELM not only brings better generalization, but also directly indicates the need for more hidden layer nodes. The increase in nodes directly leads to the increase in computer resource consumption and even fitting in the training process. In order to solve the above problems, Huang et al. [33] proposed incremental ELM (IELM). Liang et al. [34] proposed the online sequence ELM (OS-ELM), which can split and input the data into ELM, obtain the data in real time for ELM training, stabilize the generalization performance of ELM, and alleviate the pressure on ELM training due to the large amount of data in deep learning. Ganesan et al. [35] used the chimpanzee optimization algorithm to optimize the random parameters generated by ELM and achieved 95% to 98% classification accuracy on multiple coral data sets. However, there is still the problem that the algorithm falls into local optimization, and the feature redundancy caused by manually setting super parameters cannot be avoided. Due to the complexity of the living environment of marine organisms and the low resolution of underwater imaging systems, it is challenging to extract and classify image features of visual classification of marine organisms. In this context, this paper proposes a classification algorithm combining DenseNet201 neural networks [36] to extract image features and to improve ELM, which is optimized by improved FDA. The algorithm effectively improves the classification accuracy of underwater images and has been verified on the Fish4Knowledge dataset and the dataset used in URPC 2018 [37]. The main innovations of this study are as follows:

The flow direction algorithm optimized by fuzzy logic, chaotic strategy, and multi population strategy (FCMFDA) is proposed. The multi population strategy increases the diversity of individuals in the population. The application of chaos initialization gives the individuals in the population the characteristics of random distribution and stronger ergodicity and speeds up the convergence of the algorithm. The existence of fuzzy logic can better balance the exploration and development ability of algorithms.

The improved flow direction algorithm is combined with a search agent technology to map different ELM parameters with different fragments, which is used to simultaneously optimize the original randomly set weight and bias parameters, as well as the number of input nodes and hidden layer nodes in ELM so that ELM can adaptively select the appropriate network structure and obtain a better classification model.

The improved ELM algorithm (FCMFDA-ELM) is used to replace the full connection layer classifier of the conventional network for the final underwater image classification. It not only combines the ability of convolutional neural network to extract high-quality features, but also the improved ELM can adaptively select effective features and improve the classification accuracy. The performance indicators, such as classification accuracy and box graph, are used to verify the performance and stability of the algorithm proposed in this paper.

The rest of this paper is arranged as follows. Section 2 mainly introduces the relevant theoretical knowledge including ELM and FDA. Section 3 introduces the proposed FCMFDA-ELM algorithm. Section 4 is the experimental part. Conclusions are described in Section 5.

## 2. Related Work

### 2.1. Extreme Learning Machine

The extreme learning machine is a feedforward neural network. Its network structure is shown in Figure 1. In the figure, it is assumed that the number of input layer nodes is n, which is taken from the length of the feature. The number of hidden layer nodes is m,

and the number of target categories is k. Therefore, the mathematical model of the limit learning machine is:

$$y_j = \sum_{i=1}^{m} \beta_i g(W_i \cdot X_j + b_i) \; j = 1, \cdots, n \tag{1}$$

where $y_j$ is the predicted output of ELM, $g(x)$ is the activation function (a commonly used sigmoid), and $b_i$ is the hidden layer bias. $W_i = [w_{i,1}, w_{i,2}, \cdots, w_{i,n}]^T$ and $\beta_i$ are input and output weights, respectively. Formula (1) can be abbreviated as:

$$Y = H\beta \tag{2}$$

where $H$ is the output of the hidden layer:

$$H = \begin{bmatrix} g(W_i \cdot X_1 + b_1) & \cdots & g(W_m \cdot X_1 + b_m) \\ \vdots & \ddots & \vdots \\ g(W_1 \cdot X_n + b_1) & \cdots & g(W_m \cdot X_n + b_m) \end{bmatrix}_{N * L} \tag{3}$$

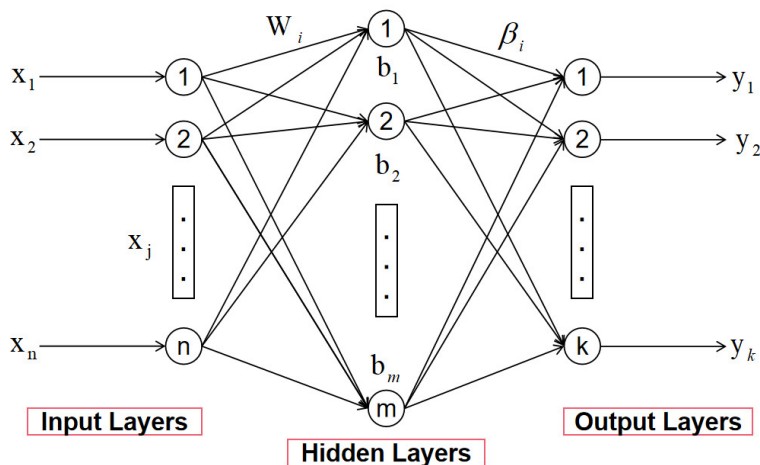

**Figure 1.** Network structure diagram of extreme learning machine.

Taking the known image features as input, combining the randomly generated weight $W_i$ and offset $b_i$ to calculate $H$, combining the known image label $Y$, and including the process of solving the output weight $\beta_i$, is the ELM training process. It is desirable to obtain $\hat{W_i}$, $\hat{b_i}$, and $\hat{\beta_i}$, such that:

$$\left\| H\left(\hat{W_i}, \hat{b_i}\right)\hat{\beta_i} - Y \right\| = \min_{W,b,\beta} \| H(W_i, b_i)\beta_i - Y \| \tag{4}$$

Formula (4) is equivalent to the minimization loss function $E$ in Formula (5):

$$E = \sum_{j=1}^{n} \left( \sum_{i=1}^{m} \beta_i g(W_i \cdot X_j + b_i) - y_i \right)^2 \tag{5}$$

Therefore, the output weight training in ELM is transformed into a linear solving problem as follow:

$$\hat{\beta} = H^+ Y \tag{6}$$

In Formula (6), $H^+$ is the Moore-Penrose generalized inverse of $H$.

### 2.2. Flow Direction Algorithm

Flow direction algorithm (FDA) [38] first creates an initial population in the basin search space, and each main stream in the population has its current position. The initialization formula is as follows:

$$flows(i) = lb + rand * (ub - lb) \tag{7}$$

where, $ub$ and $lb$ are the upper and lower bounds set at the time of population initialization, respectively, and $rand$ is a random number in (0, 1). At the same time, there are $\beta$ tributaries $neighbor$ in Formula (8) around each main stream:

$$neighbor(j) = flows(i) + randn * \Delta \tag{8}$$

where $j \in [1, \beta]$. $randn$ is a random value obeying normal distribution. $\Delta$ identifies the search range of the tributary, which decreases with the iteration. The flow speed $V$ is directly related to the gradient difference between the main stream and the tributary. The formula of the generated new main stream $newFlows(i)$ is as follows:

$$newFlows(i) = flows(i) + V * \frac{flows(i) - neighbor(j)}{\|flows(i) - neighbor(j)\|} \tag{9}$$

In addition, in FDA, when there are no tributaries with better fitness values near the main stream, FDA randomly selects another main stream, $flows(r)$. If the fitness function of this main stream is better than the current main stream, $flows(i)$, the current main stream will move toward it. Otherwise, it will move along the direction of the current main stream as in Formula (10).

$$newFlows(i) = \begin{cases} flows(i) + \overline{randn} * (flows(r) - flows(i)) & fitness(r) < fitness(i) \\ flows(i) + 2\overline{randn} * (bestFlow - flows(i)) & fitness(r) > fitness(i) \end{cases} \tag{10}$$

## 3. Underwater Image Classification Model Based on CNN and Optimized ELM

In this study, DenseNet201 is used to extract the features of underwater images, and FDA optimized by chaotic strategy, and multi population strategy is proposed. Then, the parameters optimized by improved FDA are applied to ELM by using the method of search agent mapping, and the features are input for classification.

### 3.1. Chaos Initialization

For a long time, the performance of the heuristic optimization algorithm in accuracy and convergence performance depended on the quality of the initial population to a certain extent. In most heuristic algorithms, the values of everyone in the population are randomly generated because people assume that the completely random distribution can be regarded as uniform distribution, but the experimental effect is not satisfactory. Therefore, researchers began to consider the feasibility of introducing chaos theory into population initialization.

Chaos theory is widely used in parameter optimization, chaos control, and other fields. The most remarkable feature of a series of chaotic sequence values is its high randomness and wide ergodicity. Replacing the random values in the population with chaotic variables has a greater chance of finding the expected value.

Therefore, this paper introduces chaotic initialization to optimize the initial population in FDA, to find the global optimal solution faster, and to accelerate the convergence speed of the algorithm. In order to maximize the optimization performance of chaotic initialization on FDA initial population in this paper, seven different chaotic functions (Table 1) are listed to improve the FDA algorithm, respectively, and three kinds of single-mode, multi-mode, and fixed dimensional multi-mode, with a total of nine benchmark functions (Table 2), are compared to observe the optimization effect of different chaotic functions on FDA. The experimental results are shown in Table 3.

**Table 1.** Chaotic Functions.

| Name | Map |
|---|---|
| Chebyshev map | $x_{i+1} = cos\left(i\,cos^{-1}(x_i)\right)$ |
| Iterative map | $x_{i+1} = sin\left(\frac{a\pi}{x_i}\right), a \in (0,1)$ |
| Logistic map | $x_{i+1} = ax_i(1 - x_i)$ |
| Sine map | $x_{i+1} = \frac{a}{4}sin(\pi x_i), a \in (0,4]$ |
| Singer map | $x_{i+1} = \mu\left(7.86x_i - 23.31x_i{}^2 + 28.75x_i{}^3 - 13.302875x_i{}^4\right)$ |
| Sinusoidal map | $x_{i+1} = ax_i{}^2 sin(\pi x_i)$ |
| Tent map | $x_{i+1} = \begin{cases} \frac{x_i}{0.7}, x_i < 0.7 \\ \frac{10}{3}(1 - x_i), x_i \geq 0.7 \end{cases}$ |

**Table 2.** Benchmark Functions.

| Type | Function | Range | Dim | MinValue |
|---|---|---|---|---|
| Unimodal | $f1 = \Sigma_{i=1}^{n} x_i^2$ | $[-100, 100]$ | 30 | 0 |
| | $f3 = \Sigma_{i=1}^{n}\left(\Sigma_{j-1}^{i} x_j\right)^2$ | $[-100, 100]$ | 30 | 0 |
| | $f5 = \Sigma_{i=1}^{n}\left[100\left(x_{i+1} - x_i^2\right)^2 + (x_i - 1)^2\right]$ | $[-30, 30]$ | 30 | 0 |
| Multimodal | $f9 = \Sigma_{i=1}^{n}\left[x_i^2 - 10cos(2\pi x_i) + 10\right]$ | $[-5.12, 5.12]$ | 30 | 0 |
| | $f11 = \frac{1}{4000}\Sigma_{i=1}^{n} x_i^2 - \Pi_{i=1}^{n} cos\left(\frac{x_i}{\sqrt{i}}\right) + 1$ | $[-600, 600]$ | 30 | 0 |
| | $f13 = 0.1\Big\{sin^2(3\pi x_1)$ $+\Sigma_{i=1}^{n}(x_i - 1)^2[1$ $+sin^2(3\pi x_1 + 1)\big]$ $+(x_n - 1)^2\left[1 + sin^2(2\pi x_n)\right]\Big\}$ $+\Sigma_{i=1}^{n} u(x_i, 5, 100, 4)$ | $[-50, 50]$ | 30 | 0 |
| Fixed-dimension multimodal | $f16 = 4x_1^2 - 2.1x_1^4 + \frac{1}{3}x_1^6 + x_1 x_2 - 4x_2^2 + 4x_2^4$ | $[-5, 5]$ | 2 | $-1.0316$ |
| | $f18 = [1 + (x_1 + x_2 + 1)^2(19 - 4x_1 + 3x_1^2$ $-14x_2 + 6x_1 x_2 + 3x_2^2)]$ $*[30 + (2x_1 - 3x_2)^2(18$ $-32x_1 + 12x_1^2 + 48x_2$ $-36x_1 x_2 + 27x_2^2)]$ | $[-2, 2]$ | 2 | 3 |
| | $f20 = -\Sigma_{i=1}^{4} c_i \exp\left(-\Sigma_{j=1}^{6} a_{ij}\left(x_j - p_{ij}\right)^2\right)$ | $[0, 1]$ | 6 | $-3.32$ |

**Table 3.** Fitness values of FDA optimized by seven chaotic functions on nine benchmark functions.

| | Logistic Map | Chebyshev Map | Iterative Map | Sine Map | Singer Map | Sinusoidal Map | Tent Map |
|---|---|---|---|---|---|---|---|
| **F1** | $0.00 \times 10^0$ | $1.89 \times 10^{-157}$ | $0.00 \times 10^0$ | $0.00 \times 10^0$ | $0.00 \times 10^0$ | $1.76 \times 10^{-127}$ | $0.00 \times 10^0$ |
| **F3** | $0.00 \times 10^0$ | $2.80 \times 10^{-168}$ | $0.00 \times 10^0$ | $0.00 \times 10^0$ | $0.00 \times 10^0$ | $4.61 \times 10^{-92}$ | $4.51 \times 10^{-97}$ |
| **F5** | $1.40 \times 10^1$ | $1.61 \times 10^1$ | $2.01 \times 10^1$ | $5.91 \times 10^{-7}$ | $2.23 \times 10^1$ | $2.25 \times 10^1$ | $1.81 \times 10^1$ |
| **F9** | $0.00 \times 10^0$ | $0.00 \times 10^0$ | $0.00 \times 10^0$ | $0.00 \times 10^0$ | $0.00 \times 10^0$ | $0.00 \times 10^0$ | $0.00 \times 10^0$ |
| **F11** | $0.00 \times 10^0$ | $0.00 \times 10^0$ | $0.00 \times 10^0$ | $0.00 \times 10^0$ | $0.00 \times 10^0$ | $0.00 \times 10^0$ | $0.00 \times 10^0$ |
| **F13** | $8.50 \times 10^{-22}$ | $1.10 \times 10^{-2}$ | $9.74 \times 10^{-2}$ | $1.10 \times 10^{-2}$ | $1.10 \times 10^{-2}$ | $2.10 \times 10^{-2}$ | $5.48 \times 10^{-2}$ |
| **F16** | $-1.03 \times 10^0$ | $-1.03 \times 10^0$ | $-1.03 \times 10^0$ | $-1.03 \times 10^0$ | $-1.03 \times 10^0$ | $-1.03 \times 10^0$ | $-1.03 \times 10^0$ |
| **F18** | $3.00 \times 10^0$ | $3.00 \times 10^0$ | $3.00 \times 10^0$ | $3.00 \times 10^0$ | $3.00 \times 10^0$ | $3.00 \times 10^0$ | $3.00 \times 10^0$ |
| **F20** | $-3.32 \times 10^0$ | $-3.32 \times 10^0$ | $-3.32 \times 10^0$ | $-3.32 \times 10^0$ | $-3.32 \times 10^0$ | $-3.32 \times 10^0$ | $-3.32 \times 10^0$ |

This involves ranking the fitness value of each chaotic function on each benchmark function, recording the ranking, and summing the ranking of each chaotic function. The results are shown in Table 4. The logistic map has the best overall effect on FDA's initial population optimization in all seven chaotic functions. Therefore, in this study, the logistic map is used as the chaotic function in the algorithm.

**Table 4.** Score ranking of seven chaotic functions.

|  | Logistic Map | Chebyshev Map | Iterative Map | Sine Map | Singer Map | Sinusoidal Map | Tent Map |
|---|---|---|---|---|---|---|---|
| Score ranking | 25 | 19 | 16 | 23 | 18 | 11 | 16 |

*3.2. Flow Direction Algorithm Based on Chaos Initialization and Multi Population Strategy*

STEP1: Add chaos theory to the initialization of the main stream position in Formula (7) and use logical mapping to expand the search scope of the main stream population to form a new population sequence, $flows(i + 1)$, in Formula (11):

$$flows(i + 1) = \mu \cdot flows(i) \cdot (1 - flows(i)) \tag{11}$$

STEP2: In order to further optimize the optimization capability of FDA, especially the parameter selection of ELM model, this study introduces a multi population strategy for FDA. When the initial population is copied into m sub-populations, the chaos mechanism will affect the initialization of each population. Each sub-population will evolve independently, and there is an elite population composed of individuals with the best fitness value in various populations. The most individual in the elite population is the global optimal solution, and all individuals approach the optimal solution and then jump out of the local optimal solution. Table 5 shows the pseudo code form of multi group strategy.

**Table 5.** Multi group strategy pseudo code.

| **Main loop of multi-population** |
|---|
| **For i** in number of multi-populations |
|    **If** (MaxFitness (**overall**) < MaxFitness in population(**i**)) bestFlow (**overall**) = bestFlow in population(**i**) |
|    **EndIf** **EndFor** |

*3.3. Fuzzy Logic for FDA*

The branch *neighbor* of the flow direction algorithm, under the effect of offset Δ, has better balanced its exploration and development capabilities. With the deepening of iteration, from large to small, Δ turns from large-scale exploration to optimal value development. However, when there is no better tributary in an iteration, the main stream will turn to another main stream or flow to the current optimal main stream, as shown in Formula (10). In this process, the algorithm cannot balance the ability of exploration and development, and it is easy to cause individuals to deviate from the optimal solution or fall into the local optimal. Therefore, we introduce fuzzy logic [39].

First, calculate the normalized fitness value ($NFV$) of the current main stream.

$$NFV = \frac{fitness - fitness_{\min}}{fitness_{\max} - fitness_{\min}} \tag{12}$$

where $fitness$ is the fitness value of the current mainstream and $fitness_{\max}$ and $fitness_{\min}$ are the maximum and minimum fitness values of the current population. In addition, the random vector in Formula (10) is replaced by the variable offset vector $\vec{\rho}$, where the value range of elements is [0, 2].

$$\vec{\rho}_{new} = \vec{\rho}_{old} + \Delta\rho \tag{13}$$

$$newFlows(i) = \begin{cases} flows(i) + \vec{\rho} \cdot (flows(r) - flows(i)) & fitness(r) < fitness(i) \\ flows(i) + \vec{\rho} \cdot (bestFlow - flows(i)) & fitness(r) \geq fitness(i) \end{cases} \tag{14}$$

In Formulas (13) and (14), fuzzy logic uses adaptive generation $\Delta\rho$ to update the offset vector $\rho$. Its purpose is: if the fitness value of the current mainstream individuals is low in the population, we hope that $\Delta\rho$ is non negative, which can improve their exploration ability; on the contrary, let $\Delta\rho$ be a non-positive number, and reduce the offset of $\rho$ so that the algorithm focuses on finding the optimal value in a small range. We use the membership function to fuzzy $NFV$ and $\rho$, estimate the category of $\Delta\rho$ according to the fuzzy rules in Table 6, and then defuzzification to determine the offset vector $\vec{\rho}$. The output $\Delta\rho$ of the fuzzy system uses the semantic values NE (Negative), ZE (Zero), PO (Positive). When there is no suitable tributary, the exploration and development capacity of the main stream can be balanced.

**Table 6.** Fuzzy logic rules.

| No | Rules |
|----|-------|
| 1 | NFV is Low, $\rho$ is Low $\rightarrow \Delta\rho$ is PO |
| 2 | NFV is Low, $\rho$ is Med $\rightarrow \Delta\rho$ is PO |
| 3 | NFV is Low, $\rho$ is High $\rightarrow \Delta\rho$ is ZE |
| 4 | NFV is Med, $\rho$ is Low $\rightarrow \Delta\rho$ is PO |
| 5 | NFV is Med, $\rho$ is Med $\rightarrow \Delta\rho$ is ZE |
| 6 | NFV is Med, $\rho$ is High $\rightarrow \Delta\rho$ is NE |
| 7 | NFV is High, $\rho$ is Low $\rightarrow \Delta\rho$ is ZE |
| 8 | NFV is High, $\rho$ is Med $\rightarrow \Delta\rho$ is ZE |
| 9 | NFV is High, $\rho$ is High $\rightarrow \Delta\rho$ is NE |

### 3.4. Search Agent Strategy

This paper uses the search agent technology to directly optimize the number of input nodes N and hidden layer nodes M of ELM. As the super parameters in ELM, N, and M directly determine the network structure of ELM, the internal weight matrix of ELM and the dimension of bias vector. As the two key parameters that determine the network structure, N and M cannot be modified during the operation of the network, otherwise they will lead to the mismatch of matrix dimensions in the calculation process. The common algorithm using ELM will manually set N and M and select the best configuration after repeated experiments.

The search agent mapping method presetted the maximum structure of the network, inputted the number of N input nodes and M hidden layer nodes as independent parameters into the optimization algorithm for optimization, did binary operation on the result, and mapped the result to 0 or 1 to determine whether the current node is activated. The search agent structure in this article is shown in Figure 2:

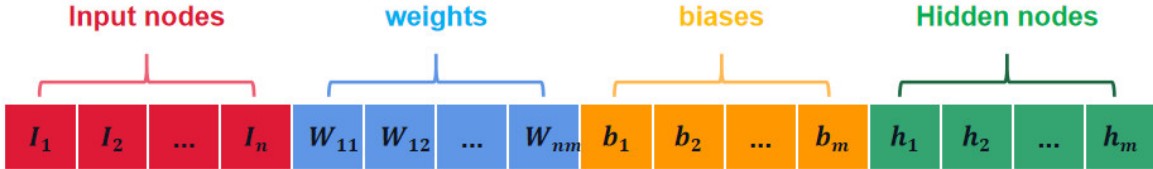

**Figure 2.** Search agent structure diagram.

Each input node and hidden layer node are regarded as independent parameters. M and N are maximized in advance and invested in iterative optimization with weight and bias in the optimization algorithm. The values of the weight and bias parameters remain unchanged, as the weight between the elm input layer and the hidden layer and the bias of the hidden layer. The parameters representing the input node and hidden layer node are binary differentiated after optimization to determine whether to activate the specified node. In this study, the search range of each particle is constrained to $[-1, 1]$. Ceil function is used to map the optimized two node parameters to 0 and 1. When the node parameter is mapped to 1, it means that this node is activated; otherwise, it means that this node is

frozen. Then, the weights and biases are rearranged according to all the activated node parameters, which can always ensure that there will be no dimension mismatch in the elm training process, and this will optimize the number of input nodes and hidden layer nodes at the same time. The detailed mapping process is shown in Figure 3:

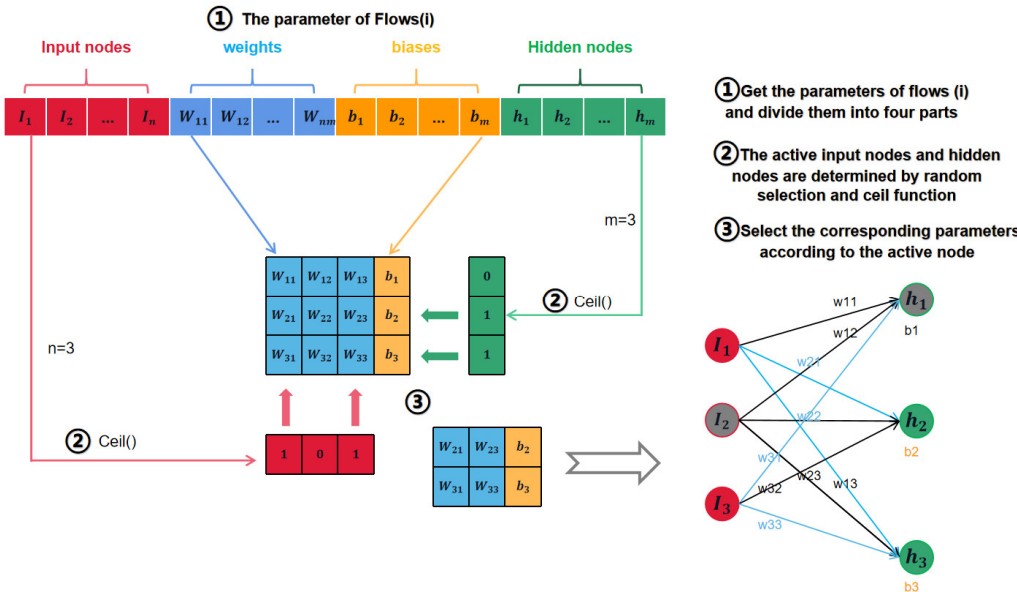

**Figure 3.** Search agent mapping flowchart.

In this paper, search agent technology is used to directly optimize the number of input nodes N and the number of hidden layer nodes M of ELM. As two key parameters that determine network structure, they cannot be modified during network operation. Common algorithms using ELM will manually set N and M and select the best configuration after repeated experiments. The maximum structure of the network is set in advance by means of searching proxy mapping, and the number of N input nodes and M hidden layer nodes are input into the optimization algorithm as independent parameters for optimization. The result is dichotomized, and the result is mapped to zero or one to determine whether the current node is activated. Combined with the multi-population strategy, the diversified population provided by the chaotic initialization strategy and the repeated iteration process, it can screen the invalid input features and optimize the number of nodes in the hidden layer to avoid the network bloat and save the tedious process of manual setting. In Figure 3, the number of input nodes and hidden layer nodes are assumed to be three to simplify the mapping process of the search agent. According to the diagram in the figure, input nodes 1 and 3 are activated, hidden layer nodes 2 and 3 are activated, and then the weight and bias matrix are cut and rearranged, which indirectly achieves the purpose of optimizing the two super parameters of input nodes and hidden layer nodes, and then the feature selection and network structure optimization are completed.

### 3.5. Flow of FCMFDA-ELM Underwater Image Classification Algorithm

The main influencing factors of elm prediction performance are: input features, the weight between input layer and hidden layer, the bias of hidden layer, the number of hidden layer nodes, and the weight between hidden layer and output layer. The weight between the hidden layer and the output layer is directly obtained by network training. In this paper, chaos initialization and multi population mechanisms are used to obtain a more random and generalized initial population. The better initial population determines the weight between the input layer and the hidden layer and the bias of the hidden layer. At the same time, the search agent mechanism is added to the optimization of FDA algorithm, which synchronously optimizes the number of input nodes and hidden layer nodes of the network, completes the selection and screening of important features and the

setting of hidden layer nodes, and improves the accuracy and stability of the classification algorithm. Set the classification accuracy of the verification set as the fitness value of the algorithm. Iterating repeatedly, as the fitness value approaches the target optimal value, the best parameter combination is selected. Finally, the flow chart of underwater image classification algorithm based on convolutional neural network, combined with FCMFDA-ELM, is shown in Figure 4. See Figure 5 for the network structure diagram. The overall operation flow of the algorithm is as follows:

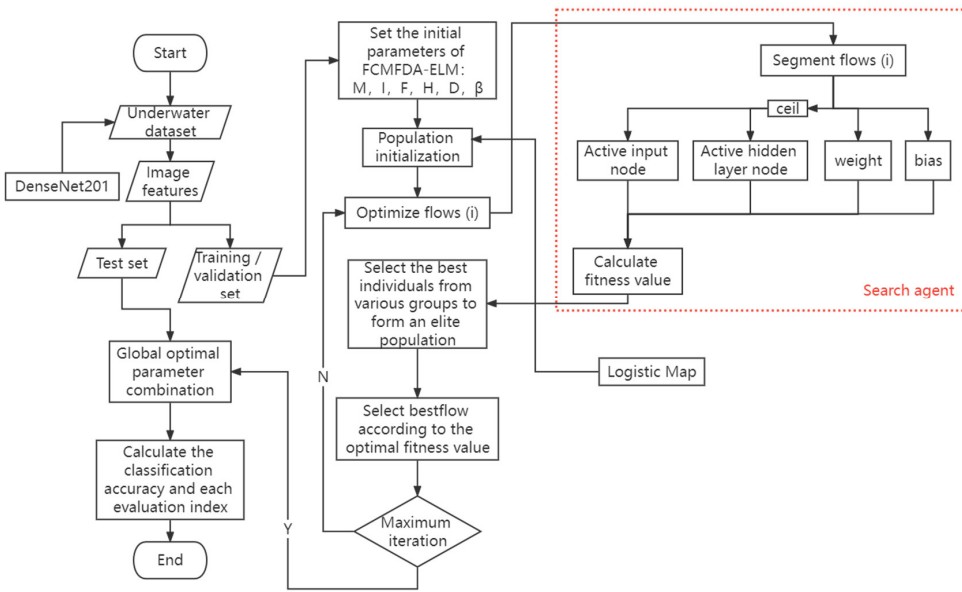

**Figure 4.** Flow chart of underwater image classification algorithm based on convolution network and ELM optimized by FCMFDA.

| Our Network |
| :---: |
| 7x7 Conv, stride 2 |
| 3x3 maxpool stride 2 |
| [1x1Conv,3x3Conv] x6 |
| 1x1Conv |
| 2x2 avgpool stride 2 |
| [1x1Conv,3x3Conv] x12 |
| 1x1Conv |
| 2x2 avgpool stride 2 |
| [1x1Conv,3x3Conv] x48 |
| 1x1Conv |
| 2x2 avgpool stride 2 |
| [1x1Conv,3x3Conv] x32 |
| 7x7 global avgpool |
| **FCMFDA-ELM** |

**Figure 5.** Structure diagram of underwater image classification network based on convolution neural network and FCMFDA-ELM.

The DenseNet201 network of ImageNet migration learning is used to extract the features of the images in the dataset and perform appropriate dimensionality reduction operations, with the category of the image as the label.

Divide the extracted data set into 8:1:1 training set, verification set, and test set. The training set is used to train the ELM network, the verification set is used to calculate the fitness value and feedback the optimization status of the network, and the test set returns the final indicators to evaluate the performance of the algorithm.

Set the initial parameters of the algorithm: the initial population number M of multiple population strategy, the number of search agents of each population, the maximum number of iterations I, the maximum number of features F, and the maximum number of hidden layer nodes H. Finally, the search agent dimension is jointly expressed as $D = F + H + (F * H) + H$ by the above parameters. Additionally, the number of tributaries β in FDA algorithm are considered.

The logistic map chaotic function is applied to the function of Equation (7) to represent the chaotic initialization of the population.

According to the way of the searching agent, the number of input nodes and hidden layer nodes will enter FCMFDA, together with weight and bias parameters, and it will start optimization according to the original FDA process. $Flows(i)$ in Equation (7) are the parameter set involved in optimization.

The result of $flows(i)$ is disassembled and binarized into four sections: activated input node, activated hidden layer node, weight parameters between input layer and hidden layer, and bias parameters of the hidden layer.

Apply the parameters of step 6 to ELM, calculate the fitness value of all $flows(i)$ on the verification set, and update the optimal fitness value of this iteration and the corresponding optimal search agent *bestFlow*. The optimization direction of other main streams shall be determined according to *bestFlow*.

Check whether the current iteration has reached the maximum number of iterations. If it has not reached the maximum number of iterations, return to step 5 and continue to optimize the main stream position in various groups. Otherwise, end the cycle and continue to the next step.

According to the final *bestFlow* obtained by the algorithm, the parameters in ELM are analyzed and applied to the test set to calculate various evaluation indicators of the algorithm and to evaluate the performance of the algorithm in this paper.

## 4. Experimental Results and Analysis

In order to analyze the performance of the underwater image classification algorithm based on convolutional neural network feature extraction and ELM optimized by FCMFDA, the experiment set up six groups of comparison methods: pure DenseNet201, FDA-ELM, STOA-ELM, WOA-ELM, MFO-ELM, and ELM. All ELM-based algorithms apply the method of search agent.

In addition, the experimental environment of this study is: CPU—AMD Ryzen 5, GPU—3600X 6-core Processor 3.80 GHz, RAM—16 GB, operating system—Windows 10, and program running platform—Matlab2018b.

### 4.1. Datasets

The Fish4Knowledge dataset contains 27,370 images of 23 kinds of underwater fish, with pixels in each image between 20 × 20 to 200 × 200. However, there are obvious differences in the number of fish images among different species. The largest category itself contains 12,112 images, while the smallest category has only 16 images. In this study, according to the number of images contained in each class, 200 images of each class in the top 10 classes of fish were selected to form a smaller dataset, as shown in Figure 6.

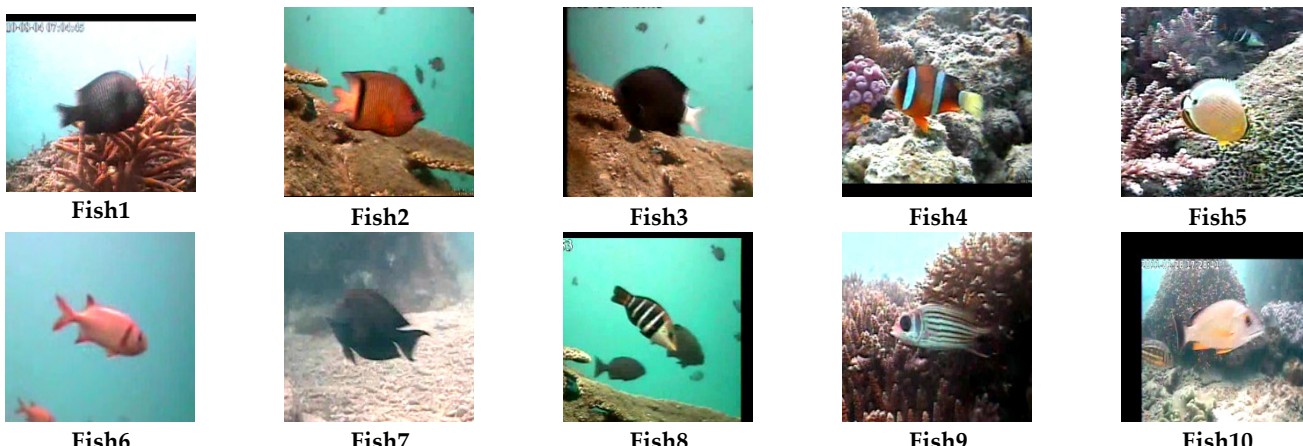

**Figure 6.** Fish4Knowledge dataset.

Underwater robot professional competition dataset, which was originally used for underwater target detection, was used. The detection targets are echinus, holothurian, scallop, starfish, and aquatic plants. There are two main files in the dataset: JPEGImages, which contains 3701 images and annotations corresponding to each image in JPEGImages. There is an identification file to record the species contained in each image and their position in the image. In this paper, all images in the URPC dataset are segmented according to the identification file in annotations to obtain the dataset for classification, as shown in Figure 7. Considering that the number of images of aquatic plants is too small, this study selects four kinds of images, except aquatic plants, for underwater image classification experiments.

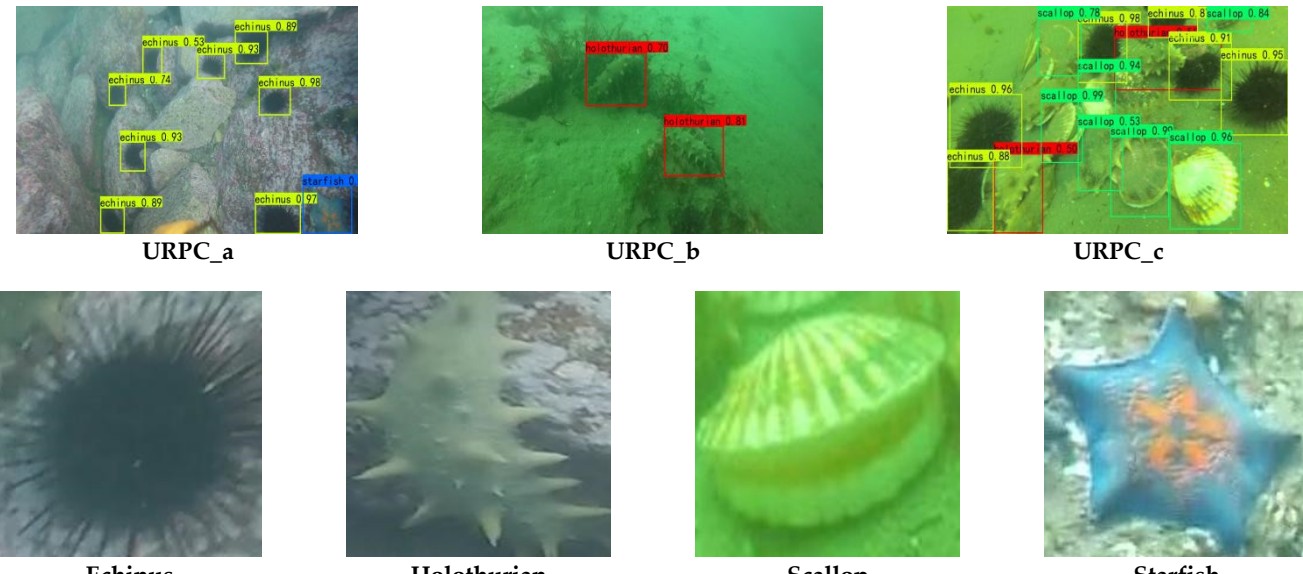

**Figure 7.** URPC dataset and new classification dataset after segmenting.

All images are uniformly scaled to 224 × 224 pixel size during the experiment. Based on the ten-fold cross validation, the experiment analyzes the experimental result data, divides all image features into 10 equal parts, takes one of them as the test set, calculates the indicator, and analyzes the performance of the algorithm.

### 4.2. Performance Evaluation Indicators

For a more rigorous analysis of the effectiveness of the proposed algorithm, this paper uses *precision*, *recall*, *accuracy*, and *F1* as the evaluation indicators of the classification algorithm. Their formulas are as follows:

$$Precision = TP/(TP + FP) * 100\% \tag{15}$$

$$Recall = TP/(TP + FN) * 100\% \tag{16}$$

$$Accuracy = TP + TN/(TP + TN + FP + FN) * 100\% \tag{17}$$

$$F1 = 2 * Precision * Recall/(Precision + Recall) * 100\% \tag{18}$$

$TP$ represents the positive case of correct prediction, $FP$ represents the positive case of wrong prediction, $TN$ represents the negative case of correct prediction, and $FN$ represents the negative case of wrong prediction. In multi-objective classification, positive examples only refer to the current class, and all classes except the current class belong to negative examples. *Precision* represents the reliability of a category identified by the algorithm, *recall* represents the probability that the algorithm model identifies a specific category in all categories, *accuracy* represents the classification accuracy of the algorithm, and *F1* is the synthesis of the first three indicators. In this experiment, the algorithm is considered as a whole, and the above indicators take the average value of multiple classes. In addition, the fitness value in the convergence process of the algorithm are calibrated as the classification accuracy of the verification set.

### 4.3. Experimental Parameter Setting

In this study, all optimization algorithms select the same size of the initial population, 20, and the number of iterations, 150. Because the large-scale numerical calculation will lead to some migration problems, the convergence curve is unstable. In order to avoid this problem, this study unifies the algorithm before optimization, and the search range of all algorithms and all population individuals is reduced to the range of [−1, 1]. In addition, for the maximum number of input nodes and hidden layer nodes set in the search agent, the optimization purpose of the number of input nodes is to adaptively select effective features. Therefore, the maximum number of input nodes is determined by the feature length extracted from the dataset, which is 587 in Fish4Knowledge and 480 in URPC. The maximum number of hidden layer nodes is uniformly set to 100, the additional population number of multiple population strategy is three, and the number of tributaries β in the FDA algorithm is one. The detailed parameters of each algorithm are shown in Table 7. In addition, the parameters of DenseNet201 network for feature extraction are set as follows: batchsize is eight, maxepoch is 10, and the initial learning rate is 0.001.

**Table 7.** Detailed parameter in each algorithm.

| Parameter | FCMFDA-ELM | FDA-ELM | STOA-ELM | WOA-ELM | MFO-ELM | ELM |
|---|---|---|---|---|---|---|
| Population size | 20 | 20 | 20 | 20 | 20 | 20 |
| Maximum number of iterations | 150 | 150 | 150 | 150 | 150 | 150 |
| Additional population | 3 | – | – | – | – | – |
| Maximum number of input nodes (Fish4/URPC) | 587/480 | 587/480 | 587/480 | 587/480 | 587/480 | 587/480 |
| Number of hidden layer nodes (Fish4/URPC) | 100 | 100 | 100 | 100 | 100 | 100 |
| Number of tributaries β | 1 | 1 | – | – | – | – |
| Particle search range | [−1, 1] | [−1, 1] | [−1, 1] | [−1, 1] | [−1, 1] | [−1, 1] |
| Logarithmic helix shape constant b | – | – | – | – | 1 | – |

*4.4. Discussion on Experimental Results*

4.4.1. Experimental Analysis on Fish4Knowledge Dataset

(1) Performance Analysis of Algorithm Classification

Figure 8 shows the confusion matrix of all algorithms involved in this study on the Fish4Knowledge dataset. Table 8 summarizes the detailed experimental results after ten-fold cross validation. The parameters n and m in the table are the optimal number of input nodes and hidden layer nodes found by the optimization algorithm and search agent, respectively. It can be seen from the table that, in the four indicators of precision, recall, accuracy and F1, the methods of using ELM as a classifier instead of the original softmax classifier in convolutional neural networks have achieved remarkable results. The indicators' value of these methods is higher than those of DenseNet201. Even the original ELM has achieved better classification results after replacing softmax classifier. In addition, the average results of FCMFDA-ELM algorithm are 0.9948 precision, 0.9949 recall, 0.9990 accuracy, and 0.9947 F1, and the four indicators' values are the highest compared with the other six algorithms.

**Table 8.** Indicators of each algorithm (Fish4Knowledge).

| Method | n | m | Norm | | | |
| --- | --- | --- | --- | --- | --- | --- |
| | | | Precision | Recall | Accuracy | F1 |
| **DenseNet201** | – | – | 0.9018 ± 0.0031 | 0.8600 ± 0.0012 | 0.9000 ± 0.0183 | 0.8263 ± 0.0312 |
| **FCMFDA-ELM** | 337 | 52 | 0.9951 ± 0.0016 | 0.9948 ± 0.0016 | 0.9990 ± 0.0001 | 0.9948 ± 0.0011 |
| **FDA-ELM** | 229 | 36 | 0.9604 ± 0.0353 | 0.9610 ± 0.0348 | 0.9606 ± 0.0346 | 0.9593 ± 0.0362 |
| **STOA-ELM** | 229 | 37 | 0.9568 ± 0.0369 | 0.9567 ± 0.0388 | 0.9567 ± 0.0381 | 0.9950 ± 0.0395 |
| **WOA-ELM** | 380 | 60 | 0.9800 ± 0.0168 | 0.9802 ± 0.0163 | 0.9801 ± 0.0152 | 0.9796 ± 0.0164 |
| **MFO-ELM** | 303 | 35 | 0.9799 ± 0.0172 | 0.9796 ± 0.0172 | 0.9797 ± 0.0158 | 0.9790 ± 0.0172 |
| **ELM** | 322 | 70 | 0.9348 ± 0.474 | 0.9355 ± 0.0475 | 0.9356 ± 0.0465 | 0.9300 ± 0.0478 |

The visual comparison of the four indicators and standard deviation of all optimization algorithms on the Fish4Knowledge dataset is shown in Figure 9. The FCMFDA-ELM classification algorithm, combined with convolutional neural network proposed in this paper, has the best indicators' values and the smallest standard deviation. Figure 10 shows the accuracy of the test set when each optimization algorithm performs ten-fold cross validation. It can be clearly seen in the figure that the classification effect of the FCMFDA-ELM algorithm, represented by a black triangular dotted line, is the best, and the fluctuation is the smallest in all ten-fold test sets.

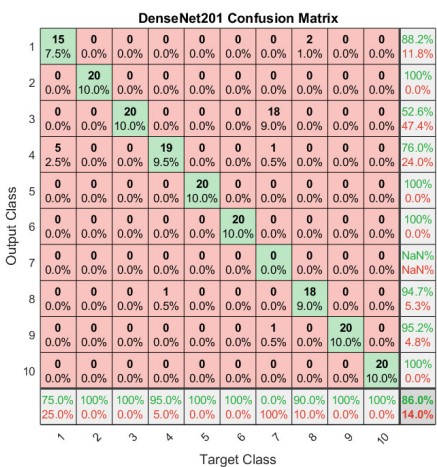

**(A) DenseNet201**　　　　　　　**(B) FCMFDA-ELM**

**Figure 8.** *Cont.*

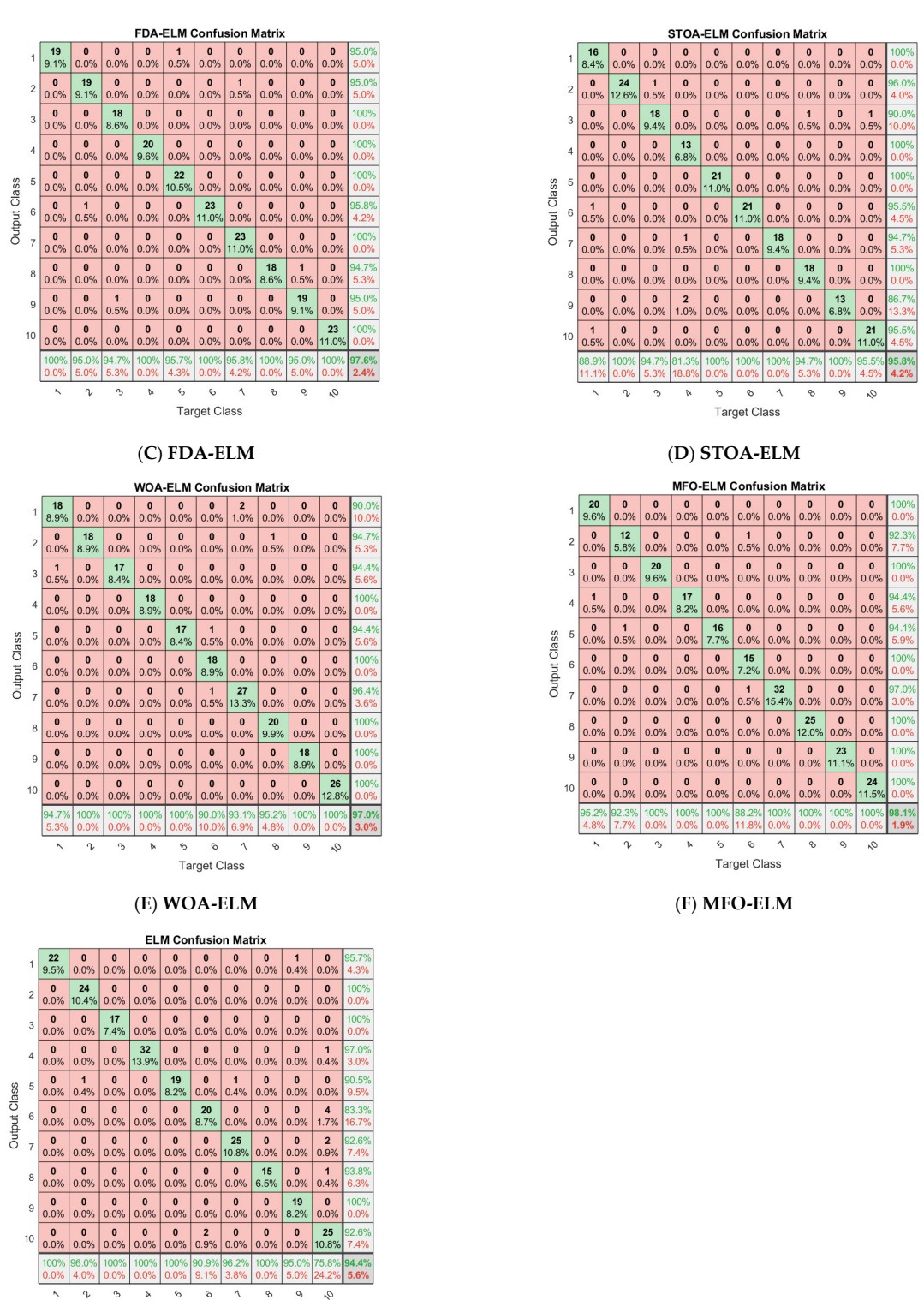

**Figure 8.** Confusion matrix (Fish4Knowledge) of each algorithm, including DenseNet201 (**A**), FCMFDA-ELM (**B**), FDA-ELM (**C**), STOA-ELM (**D**), WOA-ELM (**E**), MFO-ELM (**F**), and ELM (**G**).

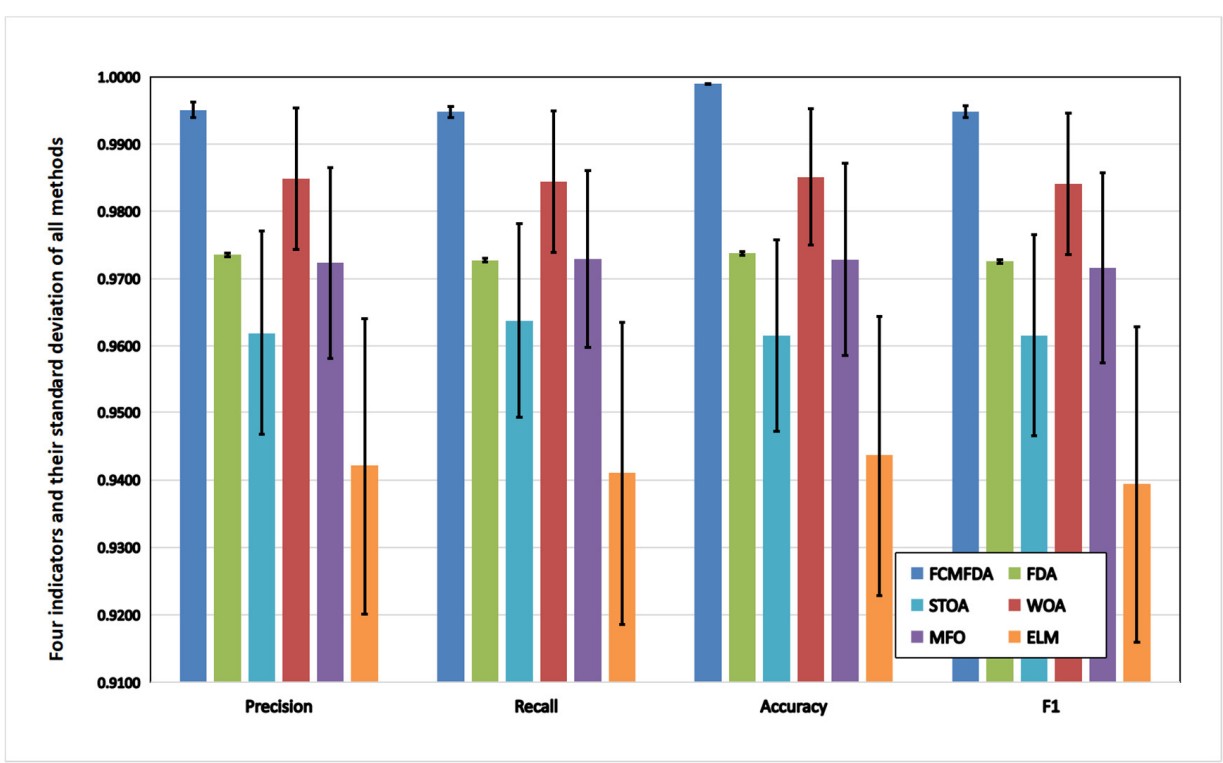

**Figure 9.** Four indicators and their standard deviation of all methods (Fish4Knowledge).

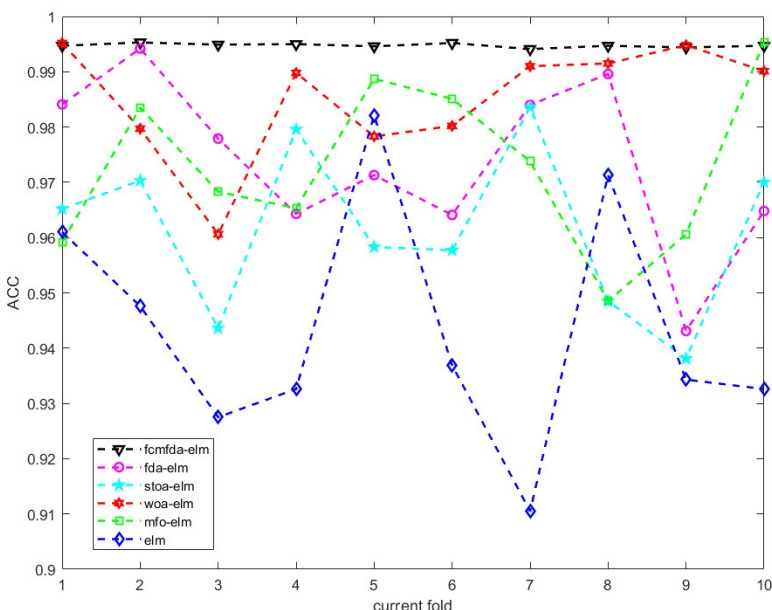

**Figure 10.** Ten-fold test set classification accuracy of each algorithm (Fish4Knowledge).

(2) Significant difference analysis

The FCMFDA-ELM classification algorithm proposed in this paper is *t*-tested on four indicators compared with other algorithms. Take the experimental results of the algorithm proposed in this paper as the benchmark and compare the experimental results of other comparison algorithms to compare whether the indicators' data of the comparison algorithm and the indicators' data of FCMFDA-ELM come from the same distribution. Taking the *p* value of *t*-test as a reference, when the *p* value is less than 0.05, it is considered that there is a distinction between the two data, and when the *p* value is less than 0.01, it is

considered that the distinction is obvious, that is, the algorithm proposed in this paper has significantly improved the classification effect. The equation for the *t*-test is as follows:

$$T = \frac{\overline{X_1} - \overline{X_2}}{\sqrt{\frac{\delta_{x_1}^2 + \delta_{x_2}^2 - 2\gamma\delta_{x_1}\delta_{x_2}}{n-1}}} \tag{19}$$

where $\overline{X_1}$, $\overline{X_2}$, and $\delta_{x_1}^2$, $\delta_{x_2}^2$ are the mean and variance of the two groups of samples, respectively, $\gamma$ is the correlation coefficient, and n is the number of samples. As can be seen from Table 9, the *t*-test results of the values of all four indicators of all other algorithms and FCMFDA-ELM are less than 0.01. This proved that the proposed algorithm has better classification performance than other algorithms.

**Table 9.** *T*-test results of FCMFDA-ELM and other algorithms on four indicators (Fish4Knowledge).

| Ours | Others | *p*-Value | | | |
|---|---|---|---|---|---|
| | | Precision | Recall | Accuracy | F1 |
| FCMFDA-ELM | FDA-ELM | $4.363 \times 10^{-4}$ | $7.011 \times 10^{-4}$ | $3.746 \times 10^{-4}$ | $5.221 \times 10^{-4}$ |
| | STOA-ELM | $4.498 \times 10^{-6}$ | $4.416 \times 10^{-6}$ | $1.832 \times 10^{-6}$ | $3.059 \times 10^{-6}$ |
| | WOA-ELM | $1.382 \times 10^{-3}$ | $8.472 \times 10^{-3}$ | $1.712 \times 10^{-2}$ | $7.732 \times 10^{-3}$ |
| | MFO-ELM | $1.571 \times 10^{-4}$ | $9.785 \times 10^{-5}$ | $2.464 \times 10^{-5}$ | $1.723 \times 10^{-4}$ |
| | ELM | $7.111 \times 10^{-8}$ | $2.724 \times 10^{-8}$ | $7.731 \times 10^{-9}$ | $4.371 \times 10^{-8}$ |

(3) Convergence analysis of algorithm

The convergence process of the ELM classification algorithm optimized by FCMFDA, FDA, STOA, WOA, and MFO on the validation set in 150 iterations is recorded in order to verify the improvement of the convergence speed of the algorithm proposed in this paper. The experimental results are shown in Figure 11. The figure shows that the FCMFDA-ELM algorithm, represented by the black triangular dotted line, has the advantages of high classification accuracy and fast convergence speed. With the help of chaotic initialization, the excellent initial population makes the FDA algorithm have a broader search space. The existence of multiple population strategy and tributaries in FDA makes the FDA population more diverse. Therefore, FCMFDA-ELM has higher convergence and basically converges to the optimal state after about 10 iterations. In addition, the value of population in ELM algorithm only depends on random generation, and there is no optimization method to provide the evolution direction of population. Therefore, the comparison of ELM algorithm is not added to the analysis of convergence.

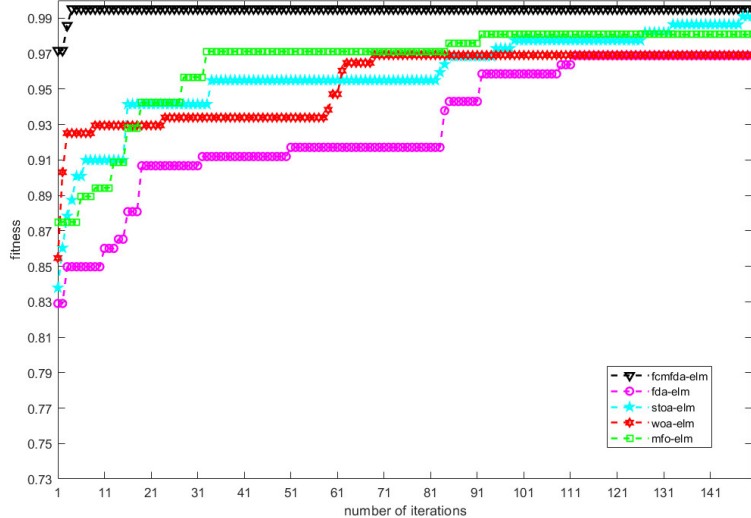

**Figure 11.** Convergence curve of each classification algorithm (Fish4Knowledge).

(4) Stability analysis of algorithm

In order to evaluate the stability of the algorithm proposed in this paper, this paper adopts the form of drawing box graph. The data source of box graph is the classification accuracy of the test set. An amount of 10 experiments are carried out on each fold in the cross validation of ten folds, and the average value is taken. The box diagram of the experimental results of each algorithm is shown in Figure 12. The upper and lower black lines in the figure represent the upper and lower boundary values of 10 data, and the upper and lower boundaries of the blue box represent the upper and lower quartiles of the data, respectively, both representing the distribution of the data. The red line represents the median of 10 data, while the red '+' represents outliers, i.e., value that deviates greatly from other values. In the box graph, the smaller the offset of the upper and lower boundaries and quartiles, the more concentrated the calculation results of the algorithm, that is, the algorithm has high stability. As can be seen from the figure, the FCMFDA-ELM algorithm in this paper not only has high classification accuracy, but also has excellent stability.

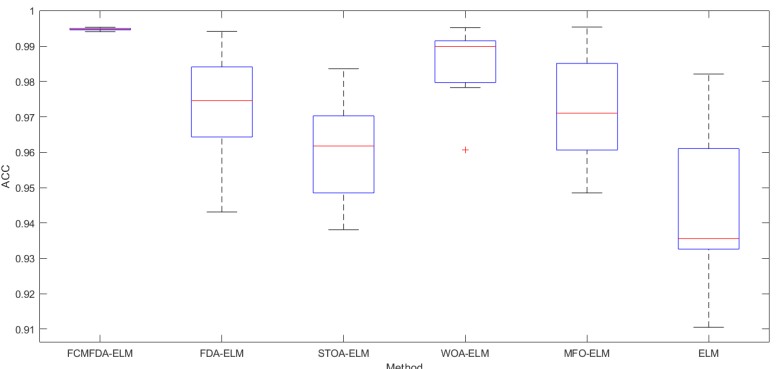

**Figure 12.** Box graph of each classification algorithm (Fish4Knowledge).

4.4.2. Experimental Analysis on URPC Dataset

(1) Performance Analysis of Algorithm Classification

Figure 13 shows the confusion matrix of all algorithms involved in this study on URPC dataset. Table 10 shows the detailed experimental results after ten-fold cross validation. It can be seen from the table that, in the values of the four indicators, the values of the algorithm using ELM and optimized ELM as classifier are also higher than those of DenseNet201, which are not different due to the change of the experimental dataset. In the URPC dataset, the results of FCMFDA-ELM classification algorithm are 0.9675 precision, 0.9637 recall, 0.9690 accuracy, and 0.9654 F1. Compared with the other six algorithms, the four indicators of FCMFDA-ELM are all the best. The performance of FCMFDA-ELM algorithm proposed in this paper is further proved.

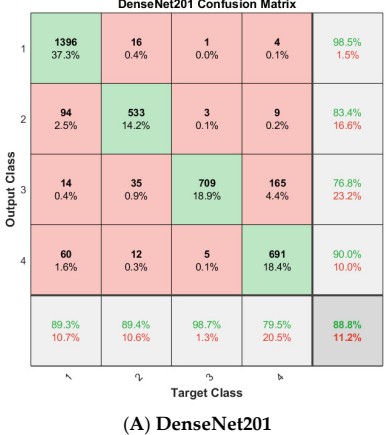

**(A) DenseNet201**

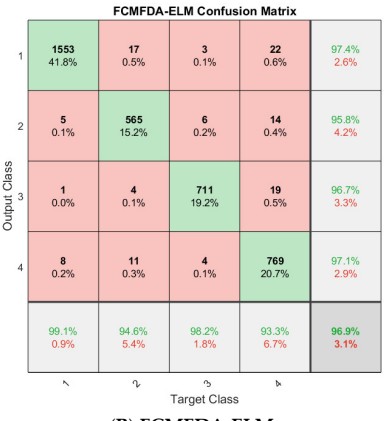

**(B) FCMFDA-ELM**

**Figure 13.** *Cont.*

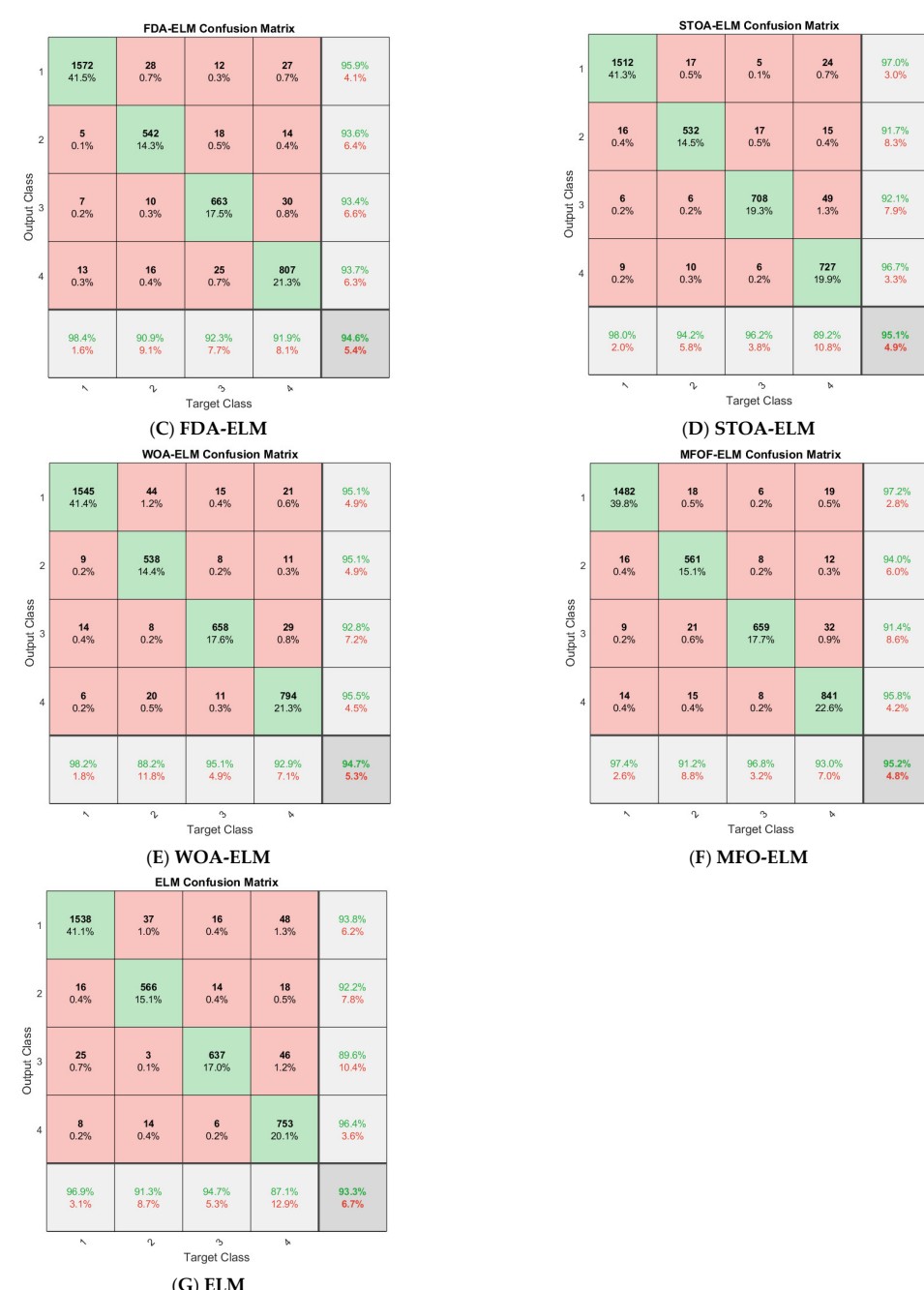

**Figure 13.** Confusion matrix (URPC) of each algorithm, including DenseNet201 (**A**), FCMFDA-ELM (**B**), FDA-ELM (**C**), STOA-ELM (**D**), WOA-ELM (**E**), MFO-ELM (**F**), and ELM (**G**).

**Table 10.** Indicators of each algorithm (URPC).

| Method | n | m | Norm | | | |
| --- | --- | --- | --- | --- | --- | --- |
| | | | Precision | Recall | Accuracy | F1 |
| **DenseNet201** | – | – | 0.8718 ± 0.0114 | 0.8924 ± 0.0086 | 0.8884 ± 0.0024 | 0.8770 ± 0.0035 |
| **FCMFDA-ELM** | 258 | 63 | 0.9682 ± 0.0050 | 0.9654 ± 0.0061 | 0.9852 ± 0.0024 | 0.9667 ± 0.0055 |
| **FDA-ELM** | 210 | 35 | 0.9309 ± 0.0156 | 0.9246 ± 0.0127 | 0.9428 ± 0.0076 | 0.9274 ± 0.0141 |
| **STOA-ELM** | 202 | 33 | 0.9505 ± 0.0090 | 0.9531 ± 0.0147 | 0.9556 ± 0.0045 | 0.9522 ± 0.0070 |
| **WOA-ELM** | 254 | 35 | 0.9419 ± 0.0080 | 0.9340 ± 0.0072 | 0.9440 ± 0.0062 | 0.9376 ± 0.0073 |
| **MFO-ELM** | 264 | 31 | 0.9366 ± 0.0240 | 0.9305 ± 0.0318 | 0.9410 ± 0.0244 | 0.9325 ± 0.0302 |
| **ELM** | 240 | 62 | 0.9316 ± 0.0199 | 0.9242 ± 0.0203 | 0.9330 ± 0.0211 | 0.9275 ± 0.0200 |

Figure 14 shows the value and the standard deviations of four indicators, which were obtained from the experiments of the algorithms used in the experiment on the URPC dataset, which is an intuitive graphical expression of Table 10. It can be clearly seen in the figure that FCMFDA-ELM classification algorithm has the best value of four indicators and the smallest standard deviation. In addition, although FDA algorithm also has a small standard deviation, it is close to the worst elm in the figure in terms of overall classification effect, which proves the effectiveness of this study on algorithm improvement. Figure 15 shows the accuracy on the test set when each algorithm performs ten-fold cross validation on the UPRC dataset. Even though the volume of URPC dataset is much larger than Fish4Knowledge dataset, the FCMFDA-ELM classification algorithm in this paper still shows a stable classification effect. It can be seen in the figure that the classification effect of FCMFDA-ELM algorithm with black triangular dotted line is the best in all ten-fold test sets, and the result fluctuation is small.

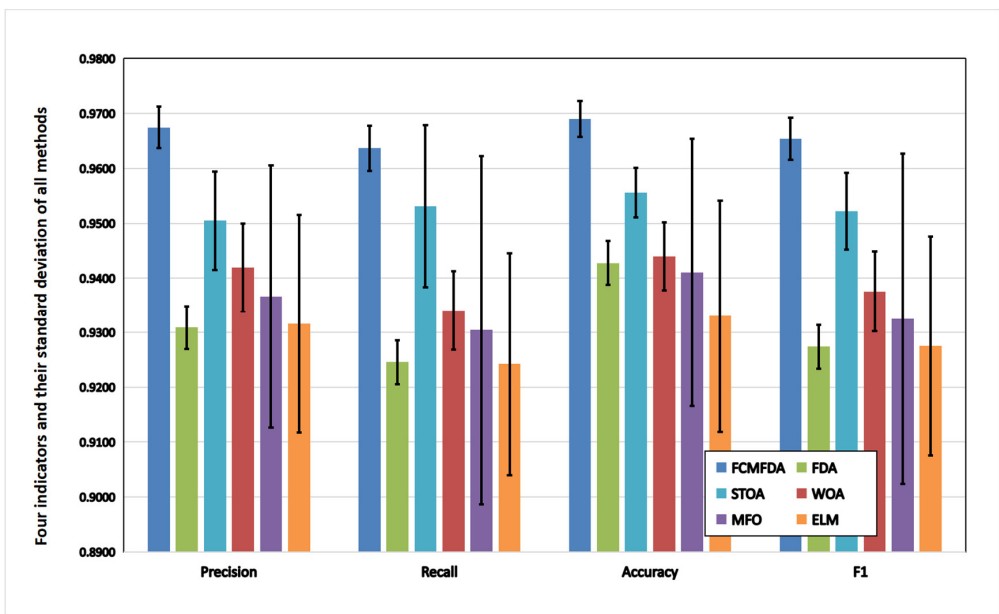

**Figure 14.** Four indicators and their standard deviation of all methods (URPC).

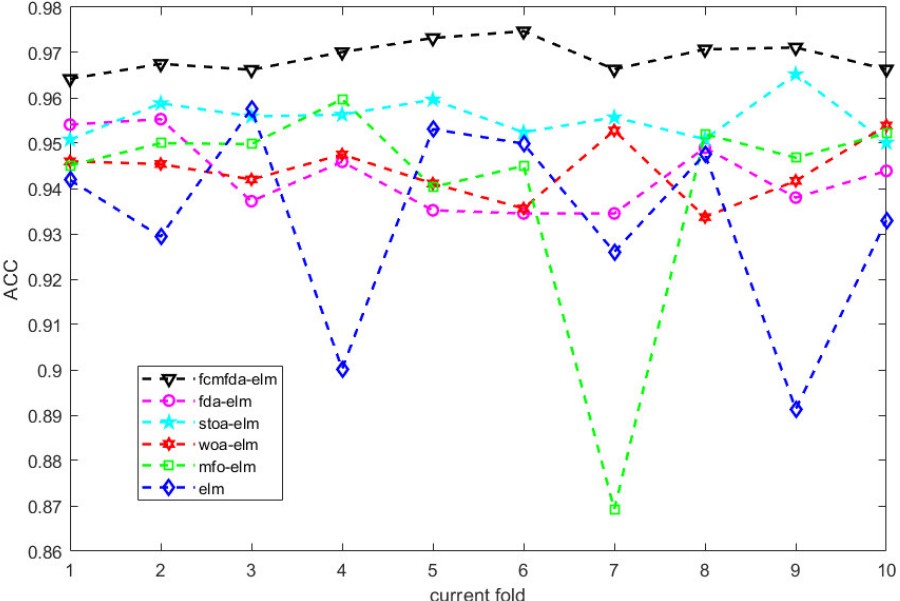

**Figure 15.** Ten-fold test set classification accuracy of each algorithm (URPC).

(2) Significant difference analysis

Table 11 records the *t*-test of the experimental results of each algorithm involved in this study on URPC dataset. The *p* value obtained by FCMFDA-ELM algorithm and other algorithms through *t*-test on four indicators basically conforms to $p < 0.01$. This excepts the *t*-test result on recall, where STOA-ELM is $4.237 \times 10^{-2}$, which is greater than 0.01, but still conforms to $p < 0.05$. It can reflect the differentiation, which is enough to prove the excellence of FCMFDA-ELM.

**Table 11.** *T*-test results of FCMFDA-ELM and other algorithms on four indicators (URPC).

| Ours | Others | *p*-Value | | | |
|---|---|---|---|---|---|
| | | Precision | Recall | Accuracy | F1 |
| FCMFDA-ELM | FDA-ELM | $2.714 \times 10^{-6}$ | $6.346 \times 10^{-8}$ | $1.251 \times 10^{-8}$ | $3.782 \times 10^{-7}$ |
| | STOA-ELM | $5.967 \times 10^{-5}$ | $4.571 \times 10^{-2}$ | $9.835 \times 10^{-7}$ | $1.234 \times 10^{-4}$ |
| | WOA-ELM | $8.351 \times 10^{-8}$ | $3.331 \times 10^{-9}$ | $3.221 \times 10^{-9}$ | $7.713 \times 10^{-9}$ |
| | MFO-ELM | $1.316 \times 10^{-3}$ | $6.263 \times 10^{-3}$ | $3.277 \times 10^{-3}$ | $4.558 \times 10^{-3}$ |
| | ELM | $4.623 \times 10^{-5}$ | $2.251 \times 10^{-5}$ | $8.313 \times 10^{-5}$ | $2.472 \times 10^{-5}$ |

(3) Convergence analysis of algorithm

Figure 16 shows the convergence of fitness values of the algorithms in this paper during training on URPC dataset. The convergence of FCMFDA-ELM and WOA-ELM in the figure is similar, and their fitness value basically converges in about 20 iterations, but FCMFDA-ELM has a better initial population, so its fitness value is about 0.04 higher than WOA-ELM. In addition, MFO-ELM, FDA-ELM, and STOA-ELM reached the optimal fitness value after 40,110, and 142 iterations, respectively. The optimization strategy proposed in this paper greatly improves the convergence speed and classification effect of the FDA algorithm.

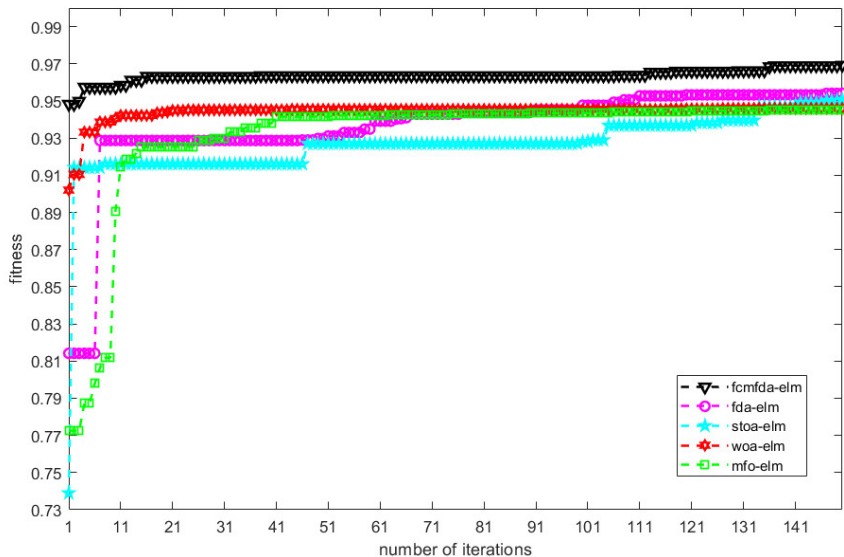

**Figure 16.** Convergence curve of each classification algorithm (URPC).

(4) Stability analysis of algorithm

In order to more rigorously analyze the stability of the algorithm proposed in this study, the box diagrams of FCMFDA-ELM, FDA-ELM, STOA-ELM, WOA-ELM, MFO-ELM, and ELM algorithms are also analyzed on the URPC dataset, as shown in Figure 17. As can be seen in the figure, FCMFDA-ELM has the highest red median line. The gap between its upper and lower boundaries and the gap between the two quartiles represented by the upper and lower boundaries of the blue box are all the smallest in the whole figure. In contrast, the ELM algorithm, which completely depends on random numerical population, has the lowest median line of classification accuracy and the largest interval

between boundaries, indicating that the results of the algorithm fluctuate greatly. The stability of other algorithms has its own advantages and disadvantages. The median line of classification accuracy of FDA algorithm without improvement is only higher than ELM, and the boundary interval is large. Therefore, it can be concluded that the improvement of FDA in this study significantly improves its classification performance and the stability of the algorithm.

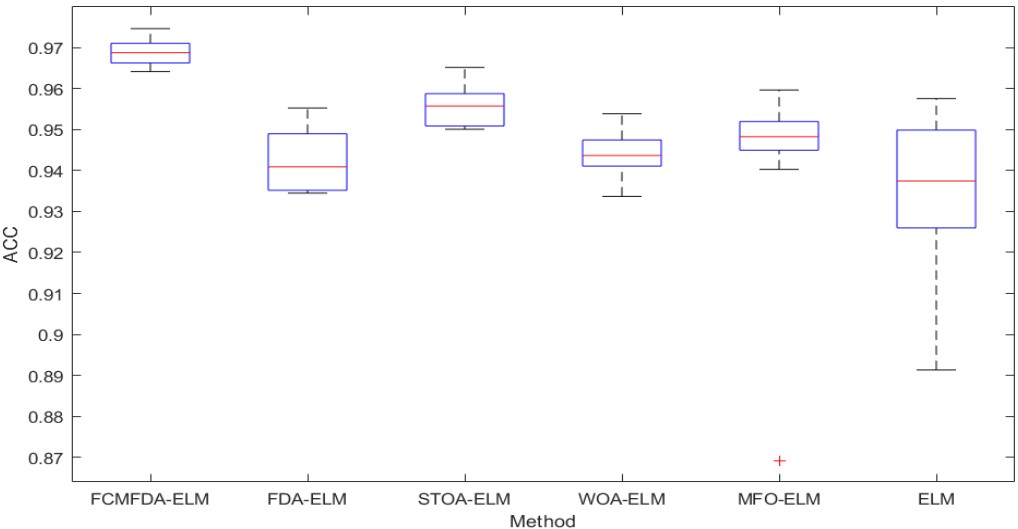

**Figure 17.** Box graph of each classification algorithm (URPC).

## 5. Conclusions

This paper proposes a new underwater image classification algorithm, which extracts features based on convolutional neural network DenseNet201, and then uses the optimized ELM (FCMFDA-ELM) to replace the softmax layer in the original convolutional neural network for underwater image classification. Based on the FDA, this study uses chaos initialization and multiple population strategy and combines the search agent method to jointly optimize ELM. The purpose is to give ELM better initialization weight and bias from input layer to hidden layer. At the same time, the number of input nodes and hidden layer nodes of ELM are added to FDA as variable parameters for optimization. Through experiments on Fish4Knowledge dataset and URPC dataset, the algorithm in this paper achieves 99.5% and 96.9% classification accuracy, respectively. Compared with the conventional full connection layer classifier in neural networks, such as DenseNet201, FCMFDA-ELM achieves 13% and 8% improvement in classification accuracy. Compared with different optimized classifiers, such as FDA-ELM, STOA-ELM, WOA-ELM, MFO-ELM, and ELM, the proposed FCMFDA-ELM has achieved 1–5% and 2–4% improvement in classification accuracy of two datasets. It is proved that, under the joint action of chaos initialization, multi group strategy and search agent mechanism, the algorithm in this paper has excellent underwater image classification accuracy, fast convergence, and strong stability.

However, the research work in this paper still has shortcomings. The cost of achieving high classification accuracy with the new FCMFDA-ELM classifier is a longer training consumption, which is worthy of further optimization. However, the trained classifier can be directly applied to the recognition network of underwater organisms, which can not only improve the classification accuracy in the recognition process, but also reduce the training burden of the recognition network. Therefore, we believe that the FCMFDA-ELM classifier proposed in this paper is still desirable.

**Author Contributions:** Conceptualization, J.Y. and M.C.; methodology, M.C.; software, X.Y.; validation, X.Y.; formal analysis, M.C.; investigation, J.Y.; resources, X.Y.; data curation, M.C.; writing—original draft preparation, X.Y. and M.C.; writing—review and editing, J.Y. and Z.Z.; visualization, X.Y.; supervision, Z.Z.; project administration, J.Y. All authors have read and agreed to the published version of the manuscript.

**Funding:** This work is supported by the National Key R&D Program of China (No. 2022YFC2803903) and the Key R&D Program of Zhejiang Province (No. 2021C03013).

**Institutional Review Board Statement:** Not applicable.

**Informed Consent Statement:** Not applicable.

**Data Availability Statement:** Fish4Knowledge dataset: (https://github.com/Callmewuxin/fish4 konwledge); URPC Dataset: (https://github.com/yanzhuangzhuang-beep/voc-URPC).

**Acknowledgments:** We appreciate the support of the National Key R&D Program of China (No. 2022YFC2803903) and the Key R&D Program of Zhejiang Province (No. 2021C03013).

**Conflicts of Interest:** The authors declared no potential conflict of interest with respect to the research, authorship, and/or publication of this article.

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
