# Peer review of "Underwater Image Classification Algorithm Based on Convolutional Neural Network and Optimized Extreme Learning Machine"

_jmse, doi:10.3390/jmse10121841_

Round 1

Reviewer 1 Report

This study proposes the flow direction algorithm (FDA) and the search agent strategy for underwater target classification. The extreme learning machine (ELM) strategy optimizes weight, bias, and super parameters.

The manuscript is well written, with good technical support. I found a minor detail in the manuscript number sections. Also, it is recommended to improve the result's discussion in the conclusion. I suggest a quantitative analysis description. 

Another important issue is the comparative analysis with other state-of-art algorithms. In particular, the following studies of underwater image classification also propose the extreme learning machine. My recommendation is to extend the discussion related to the advantages/disadvantages and differences with the published studies in https://doi.org/10.1155/2018/1214301,  https://doi.org/10.1155/2020/6707328, and https://doi.org/10.3390/s18051490.

Reviewer 2 Report

Please read the attachment. thank you.

Round 2

Reviewer 1 Report

The authors have addressed all my recommendations. I don't have more comments/suggestions, just to congrats the authors.

Reviewer 2 Report

Dear Authors and Editors: 

The authors have answered and corrected all my questions and comments carefully. The reviewer suggests that the manuscript should be accepted for publication. 

Thank you. 

Best regard, 

Reviewer.